# Perceptual visual dependence for spatial orientation in patients with schizophrenia

**Rima Abdul Razzak**[1]*, **Haitham Jahrami**[2,3], **Mariwan Husni**[3,4], **Maryam Ebrahim Ali**[2], **Jeff Bagust**[5]

**1** Department of Physiology, College of Medicine and Medical Sciences (CMMS), Arabian Gulf University (AGU), Manama, Bahrain, **2** Ministry of Health (MOH), Manama, Bahrain, **3** Department of Psychiatry, College of Medicine and Medical Sciences (CMMS), Arabian Gulf University (AGU), Manama, Bahrain, **4** Northern Ontario School of Medicine University, Ontario, ON, Canada, **5** Faculty of Health and Social Sciences, Bournemouth University, Poole, United Kingdom

* reemala@agu.edu.bh

**Data Availability Statement:** All relevant data are within the article and its Supporting Information files.

**Funding:** The author(s) received no specific funding for this work.

## Abstract

### Background

Patients with schizophrenia are reported to have vestibular dysfunction and to weigh vestibular input to a lesser extent compared to healthy controls. Such deficits may increase visual dependence (VD) for spatial orientation at a perceptual level in these patients. The aim of this study is to compare VD levels between healthy control and patients with schizophrenia and to explore associations between VD and clinical measures in these patients. Relation of VD to antipsychotic drug treatment is also discussed.

### Method

18 patients with schizophrenia and 19 healthy controls participated in this study. The Rod and Disc Test (RDT) was used to create an optokinetic surround around a centrally located rod. Participants aligned the rod to their subjective visual vertical (SVV) in both static and dynamic disc conditions. VD was calculated as the difference in SVV between these two conditions.

### Results

There was no group difference or gender difference in static or dynamic SVV as well as VD. There was no correlation between VD and any of the Positive and Negative Syndrome Scale (PANSS) scores, however VD was significantly correlated to illness duration in the patient group.

### Conclusions

Schizophrenia is not associated with greater VD levels at a perceptual level, compared to controls, indicating adequate visuo-vestibular integration for judging line verticality in these patients. Patients with greater chronicity of the disease are more visually dependent than those less chronically ill, consistent with previous reports of possible vestibular dysfunction

**Competing interests:** The authors have declared that no competing interests exist.

in patients with schizophrenia. This may affect their daily functioning in dynamic visual environments.

## Introduction

Visual dependence (VD) is a term used to describe the extent to which an individual relies on visual cues for spatial orientation [1,2]. Individuals with VD have difficulty in resolving situations wherein visual information is complex or inaccurate. Recent studies have reported that VD is considered a form of sensory reweighting deficit, and that people with VD are unable to flexibly reduce the weighting of inaccurate visual input while increasing the weighting of input from the proprioceptive or vestibular systems to compensate for this deficit. The increased visual dependence is often associated with a greater risk for falls as individuals become unable to reduce their reliance on visual feedback, even when vision is unreliable [3,4]. Visual dependence at a perceptual level can be measured by the rod and disc test (RDT) [5–7]. It is one of the most frequently employed designs used to measure the effect of visual motion on gravitational verticality judgments, or the subjective visual vertical (SVV). SVV is an indicator of spatial orientation without any vertical reference and requires multisensory integration of vestibular, visual and proprioceptive inputs [8,9]. Its coherence is crucial for appropriate postural control.

In the RDT, participants initially align a rod to what they perceive to be gravitational vertical against a stationary disc background with no cues to verticality (Static subjective visual vertical, or static SVV). They then repeat the task against a background of optokinetic disc roll motion; this has been termed dynamic subjective visual vertical (SVV) [6]. The effect of the roll motion is to bias estimates of verticality in the direction of motion and induce ocular torsion [10]. The difference in tilt between the static SVV and dynamic SVV conditions is referred to as visual dependency (VD), i.e., the change in tilt associated with a moving visual background compared with a static baseline. It has been reported that with increased VD, rotation of a visual surround can induce vection, or the sense of self-rotation in stationary observers [11–13].

Vestibular information is the primary sensory input counteracting disorienting visual or somatosensory stimuli affecting verticality perception [6]. Patients with schizophrenia (SCZ) have been reported to suffer from impairments in vestibular function [14,15] and multisensory integration impairments [16]. Additionally, vestibular sense, an internally generated reference for gravitational vertical [17], might be less weighed in patients with schizophrenia, due to the generalized reduced weighting of self-generated interoceptive signals to guide perception in these patients [18].

Given the aforementioned evidence of vestibular deficits in schizophrenia as well as the importance of vestibular information in multisensory integration for spatial orientation, we hypothesize that increased visual dependence would be present in individuals with schizophrenia, manifested as increased dynamic SVV error on the rod and disc test. This deficit may manifest as impaired functioning and spatial disorientation in dynamic visual surroundings with increased visual motion and optic flow, such as crowded or busy environments, as walking in supermarket aisles, movements of crowds or traffic, moving images at the cinema, trees swaying, or while driving fast on a highway.

This study aims to investigate the effect of a rotating background visual scene (disc) in producing a distortion of the apparent verticality of a static rod in patients with schizophrenia. Another aim is to explore whether there may be any associations between VD and severity of

clinical symptoms in these patients. This may provide further insight into multisensory integration for spatial orientation in these patients.

## Materials and methods

### Participants

Nineteen healthy control subjects (5 females, 26.3%) with no history of mental illness, neurological, or ophthalmic disorders were recruited among the staff from the same psychiatry hospital. These healthy individuals were taken into a control group. No control subject was receiving any regular medications.

Eighteen psychopathologically stable patients (5 females, 27.8%) who met the Diagnostic and Statistical Manual of Mental Disorders 5[th] Edition (DSM-5) criteria for schizophrenia were recruited from the national centre for diagnosis and treatment of severe mental illnesses in Bahrain. All diagnoses were made by a multi-disciplinary team led by a psychiatry consultant in Bahrain.

At the time of the study, all SCZ were medicated with only one patient on typical antipsychotics. For the other patients, (83.3% on atypical antipsychotic drugs, 22.2% on combination antipsychotic drugs, 27.8% on typical antidepressants, and 16.7% taking both). None of the patients were on atypical or combination ant-depressant drugs, or on lithium and propranolol tranquillizers. Clinical symptoms were assessed with the Positive and Negative Syndrome Scale (PANSS) by well-trained psychiatrists. All SCZ patients had both negative and positive symptoms, and experienced auditory hallucinations.

Exclusion criteria include important comorbid psychiatric disorder, neurological or medical disorders, severe visual loss, history of severe head trauma, alcohol/substance dependence or abuse, electro-convulsive therapy in recently initiated treatment for SCZ. All participants had normal or corrected-to-normal visual acuity when tested with Snellen charts, and subjects wore their normal corrective spectacles if necessary, during testing.

This study was approved by the Secondary Health Care Research Committee (SHCRC) at the Ministry of Health (MOH) and form the Research and Ethics Committee at the College of Medicine and Medical Sciences at the Arabian Gulf University in Bahrain (Reference: E23-PI-01/20).

After the aim and impact of the study were explained to the healthy controls and to the patients and their relatives, informed verbal consent was taken. An impartial witness, who was not a member of the study team and worked at the center where the patients were treated, endorsed that the consent from patients and their relatives was voluntary and freely given. Only those who volunteered were included in the study. Table 1 displays demographic characteristics for participants and clinical characteristics for the patient group.

### The computerized rod and disc test (CRDT)

**Measurement of SVV.**   We utilized a computerized version of the rod and disc test (CRDT) to assess verticality judgement. A virtual line marked by five white dots was used instead of a continuous line to reduce clues to verticality, which might be provided by the stepped appearance of a displayed solid line [19]. The test was performed while sitting in a comfortable position with no head restraint; however, participants were instructed to keep their trunks and heads fixed. Subjects observed the screen while sitting upright on an armless chair while their knees were extended and their feet were dorsiflexed so that only the heels were in contact with the floor in order to diminish possible proprioceptive cues from the plantar surface of the feet. The screen was placed at eye level at a distance of 80 cm, providing a full-field stimulus [7,20]. A round black paper ring was stuck on the laptop screen to conceal its edges and reduce clues to verticality, while exposing the rod and disc presentation in the

**Table 1. Demographic characteristics for all participants, with clinical characteristics for patients with schizophrenia.**

|  | Control (n = 19) | SCZ (n = 18) |  |
|---|---|---|---|
| **Age (Years)** | 32.8 ± 9.7 (20.0–53.0) | 35.6 ± 15.0 (18.0–67.0) | t = 0.67, P = 0.51 |
| **College Education level** Primary Secondary BSc Postgraduate | 2 5 8 2 | 0 10 5 0 | Chi-Square = 6.59 P = 0.09 |
| **Age at diagnosis of disease (Years)** |  | 21.2 ± 4.3 (15.0–28.0) |  |
| **Duration of illness (Years)** |  | 14.4 ± 11.8 (0.1–39.0) |  |
| **Inpatient Stay Duration (Months)** |  | 3.8 ± 2.4 (1.0–8.0) |  |
| **Scores on Psychiatry Tests** PANSS–N PANSS–P PANSS—GP |  | 24.3 ± 7.2 (9.0–32.0) 20.2 ± 6.7 (11.0–31.0) 29.1 ± 8.1 (19.0–41.0) |  |

PANNS: The Positive and Negative Syndrome Scale. "N" for negative symptoms, "P" for positive symptoms, "GP" for general psychopathology subscale.

center of the screen (Fig 1). The test was performed in a dark room minimizing further any vertical cues within the room.

Participants rotated the dots around their virtual center in 0.5˚ increments in either clockwise (CW) or counter clockwise (CCW) directions using the mouse buttons until the "rod" was considered vertical. The space bar of the computer keyboard was then pressed to record the rod alignment relative to vertical and move the program to the next presentation. As shown in Fig 2, recording of rod alignment tilt was conducted with 12 presentations in total for the 3-disc contexts with four presentations for each one: the disc static (0˚, $Disc^{0˚/s}$) for measurement of static SVV; the disc rotating to the right at an angular velocity of 30˚/s (+30˚/s, $Disc^{+30˚/s}$) or to the left (-30˚/s, $Disc^{-30˚/s}$) for measurement of dynamic SVV. At the beginning if the test, two practice trials were introduced to ensure participants understood the task at hand. These were not included in the data analysis.

We have utilized an angular velocity for the disc of 30 degrees/sec because it induces the maximum amount of tilt, which remains approximately constant when increasing the velocity further [6].

Only four trials were completed for each rotating disc condition because it is quite disorientating, so trial number was kept to a minimum. The order of display presentations was randomly selected by the computer. Participants were informed of the importance of spatial accuracy, and that the trials were not time restricted.

## CRDT measures

The angular deviation of the rod's final position from true vertical was recorded as error in degrees. According to convention, CW tilts of the rod by the participants were denoted by a positive value, whereas CCW tilts were considered negative. Unsigned (absolute) errors of the static SVV, dynamic SVV with clockwise and counter clockwise background stimulus motion were used for analysis in this study. Even though participants were advised for spatial accuracy, the time for rod alignment in each trial was also recorded by the software.

## Statistical analysis

Statistical analysis was performed by version 28 of the *Statistical Package for the Social Sciences (*SPSS) software (IBM, Chicago, USA). All data followed a normal distribution when evaluated

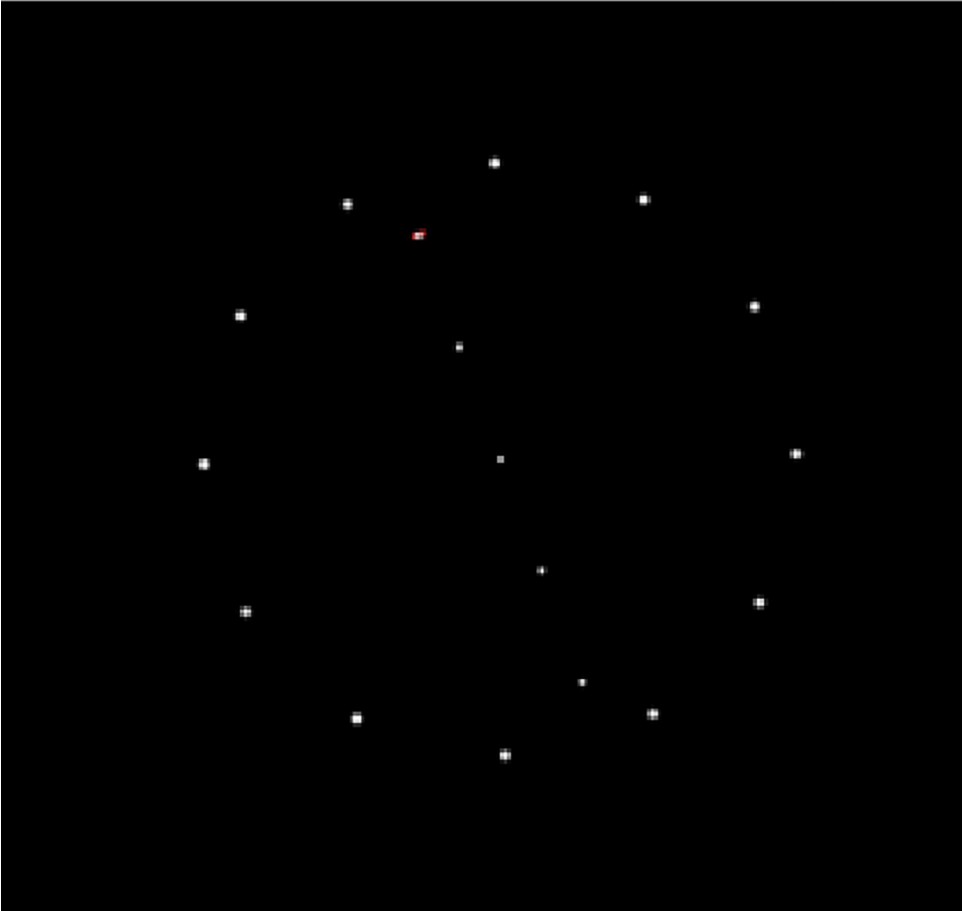

**Fig 1. Presentations of "rod and disc" during testing.** Some presentations were with no surrounding disc. With the disc presentations, the disc was either static, or rotating clockwise or counter clockwise at +30˚/s or 30˚/s respectively. The order of presentations was randomly assigned by the computer.

with the Kolmogrov and Smirnov test, so parametric tests were used for analyses. A Two-Way mixed ANOVA was used to determine the effect of psychotic condition (Control, Patient) and disc condition (Static, CCW Dynamic, CW Dynamic) on SVV errors and on time for rod alignment in each trial. Also, a Two-Way ANOVA was conducted to determine the effect of psychotic condition (Control, Patient) and gender (Male, Female) on visual dependence (VD) level.

## Results and discussion

Table 2 displays the absolute SVV errors in static and dynamic conditions of the background disc for controls and patients with schizophrenia. Analysis by the Two-Way Mixed ANOVA on the effects of psychotic condition and disc condition showed that the assumption of sphericity was violated, as assessed by Mauchly's Test of Sphericity ($P < 0.001$). Therefore, degrees of freedom were corrected using Greenhouse- Geisser estimates of sphericity ($\varepsilon = 0.58$). There was a significant main effect of disc condition on SVV errors F $(1.41, 35) = 5.22$, $P = 0.02$, partial $\eta^2 = 0.13$. SVV errors in the dynamic disc conditions (mean: CCW dynamic = 2.02; CW dynamic = 1.74) were of greater magnitude than those for the static condition (mean = 1.31). There was no significant difference between CCW dynamic SVV and CW dynamic SVV.

**Fig 2. Box and Whisker plots with median and inter-quartile range of visual dependence (VD) level data.** "X" represents the mean of the data. n = 19 for controls an n = 18 for patients with schizophrenia.

There was no significant main effect of psychotic condition F (1, 35) = 0.86, P = 0.36, partial $\eta^2$ = 0.02, on SVV errors, with controls (mean = 1.53) and patients (mean = 1.86) performing similarly overall. There was no significant interaction between psychotic condition and disc condition F (1.41, 35) = 2.40, P = 0.12, partial $\eta^2$ = 0.06, indicating that there was no group difference in SVV for any disc condition (Table 2).

**Table 2. Absolute (unsigned) deviation errors of SVV in static and dynamic conditions of the background disc for controls and patients with schizophrenia.**

|  | Controls | Patients |
|---|---|---|
| Static SVV˚ | 1.19 ± 0.81 (0.13–3.63) | 1.53 ± 0.99 (0.38–3.63) |
| CCW Dynamic SVV˚ | 1.60 ± 1.08 (0.18–3.50) | 2.45 ± 2.20 (0.38–8.38) |
| CW Dynamic SVV˚ | 1.80 ± 1.49 (0.50–6.50) | 1.68 ± 0.97 (0.38–3.75) |
| Mean Dynamic SVV˚ | 1.70 ± 0.96 (0.44–4.1 | 2.07 ± 1.47 (0.38–5.31) |

Controls (n = 19) and Patients (n = 18). Static refers to a non-moving background disc, while dynamic refers to a rotating disc at 30˚/s in either clockwise or counter clockwise direction.

Since there was no significant difference between CW and CCW absolute dynamic SVV for any of the groups, the mean of their absolute values was used to represent dynamic SVV for VD calculation. VD level was calculated as the mean of absolute values of the rod tilt from each trial of dynamic SVV minus the mean absolute static SVV rod tilt. Fig 2 is a Box and Whisker plot of VD data for both groups.

The two-way ANOVA with psychotic condition (Control, Patient) and gender (Male, Female) as independent factors revealed that there was no significant main effect of psychotic condition F (1, 33) = 0.46, $P$ = 0.50, partial $\eta^2$ = 0.014, on visual dependence level, with controls (mean = 0.51˚) and patients (mean = 0.63˚) performing similarly overall. The same applies for gender main effect F (1, 33) = 0.92, $P$ = 0.35, partial $\eta^2$ = 0.027, with males (mean = 0.50˚) displaying comparable levels of visual dependence as females (0.76˚). The interaction between psychotic condition and gender was not significant either (F (1, 33) = 0.28, $P$ = 0.60, partial $\eta^2$ = 0.008).

As for time of rod alignment, results from the Two-Way mixed ANOVA on the effect of psychotic condition (Control, Patient) and Disc condition (Static, CCW Dynamic, CW Dynamic) on rod alignment time show violation of assumption of sphericity ($P$ < 0.05), and degrees of freedom were corrected using Greenhouse- Geisser estimates of sphericity ($\varepsilon$ = 0.81). There was no significant main effect of disc condition on rod alignment time F (1.67, 35) = 0.14, $P$ = 0.83, partial $\eta^2$ = 0.004. Alignment times in the dynamic disc conditions (mean: CCW dynamic = 11.01s; CW dynamic = 10.59s) were similar to the static condition (mean = 10.74s). There was no significant main effect of psychotic condition F (1, 35) = 2.85, $P$ = 0.11, partial $\eta^2$ = 0.075, on alignment time, even though patients (mean = 11.94s) aligned the rod slower by 2.26 s than controls (mean = 9.68s). The interaction between psychotic condition and disc condition was not significant either, F (1.67, 35) = 0.035, $P$ = 0.95, partial $\eta^2$ = 0.001, indicating that there was no group difference in alignment time for any disc condition.

Even though there was no group difference in VD level, we explored through linear regression analyses whether VD level in the patients with schizophrenia was associated with severity of symptoms as measured by PANSS scores, or with illness duration. There was no correlation between any PANSS score and VD level (PANSS-N: r = 0.03, $P$ = 0.91; PANSS-P: r = 0.05, $P$ = 0.85, PANSS-GP: r = 0.06, $P$ = 0.82). VD level was however significantly correlated with illness duration (r = 0.69, $P$ = 0.001) (Fig 3).

To ensure that correlation of VD with illness duration was not a result of ageing, we included linear regression analysis between VD and age for both groups, (Fig 4). There was a significant correlation between the two factors for the patient group (r = 0.74, $P$ = 0.0004), but not for the control group (r = - 0.30, $P$ = 0.21).

This is the first study to investigate perceptual visual dependence in patients with schizophrenia while using a background optic flow that produces a distortion of the apparent verticality of a static stimulus. Visual dependence can result in signs, such as measurable imbalance and an inherent sense of instability [21]. and is associated with other symptoms or part of syndromes, including vestibular diseases, anxiety, motion sickness, and migraine.

In the present study, we tested visual dependence at a perceptual level in patients with schizophrenia by the rod and disc test (RDT), a manipulation known to bias estimates of verticality in the direction of motion in healthy participants. Higher visual dependence is thought to be a compensatory response to vestibular or proprioceptive impairments [22]. Based on the idea that patients with schizophrenia have vestibular deficits and favour exteroceptive signals to guide perception [18,23], we expected a greater influence of the dynamic visual surround on verticality perception in patients with schizophrenia compared to healthy controls.

Contrary to our expectations, patients with schizophrenia manifested similar biases in dynamic SVV as healthy controls, as they adjusted the rod as accurately as controls, indicating

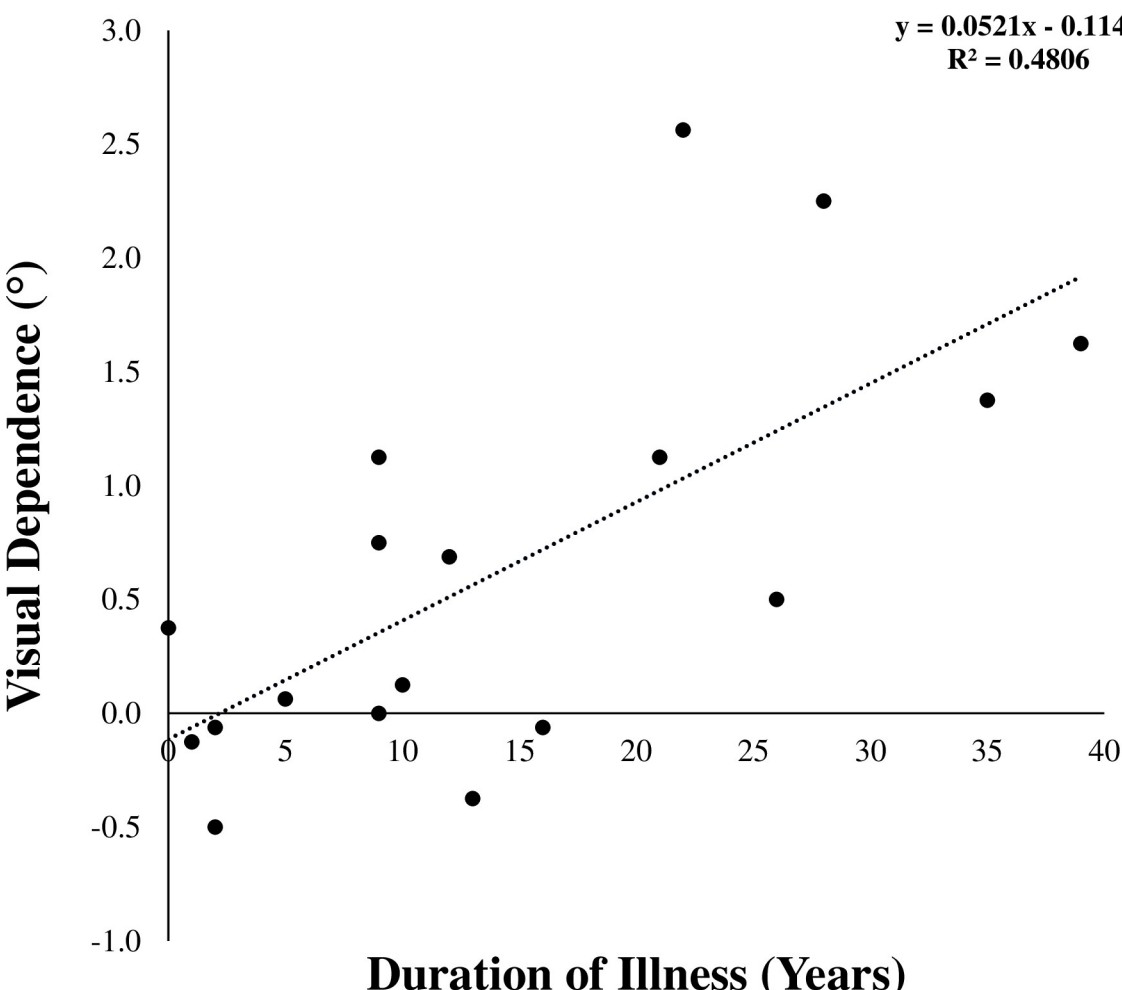

**Fig 3. Visual dependence level in patients with schizophrenia as function of illness duration.** Negative values for VD indicate that static SVV was greater than dynamic SVV values.

they were similarly affected by the background roll motion. This finding suggests that patients show a normal ability to combine exteroceptive visual cues with interoceptive vestibular information to judge the verticality of a static rod and supports the hypothesis of a weighted multi-sensory integration when estimating direction of gravity with optokinetic stimulation. These results do not support the hypothesis that patients with schizophrenia use preferentially exteroceptive visual over vestibular cues for spatial orientation in dynamic visual environments.

Our results are consistent with those in a similar recent study on visual illusion by Seymor and Kaliuzhna [23] who utilized the Tilt Illusion to probe the respective weight given to visual and vestibular cues in judging line orientation in patients with schizophrenia. This illusion does not entail a dynamic surround, but the orientation in its surround biases the perceived orientation of a vertical grating. These authors reported comparable Tilt Illusion magnitudes in healthy controls and patients during both upright and tilted head conditions. A major difference between our study design and that of the Tilt Illusion is that the moving visual displays in our study entailed no cues to visual orientation that could conflict with vestibular cues of gravity and that the surrounding motion only alters the subjective vertical by modulation of

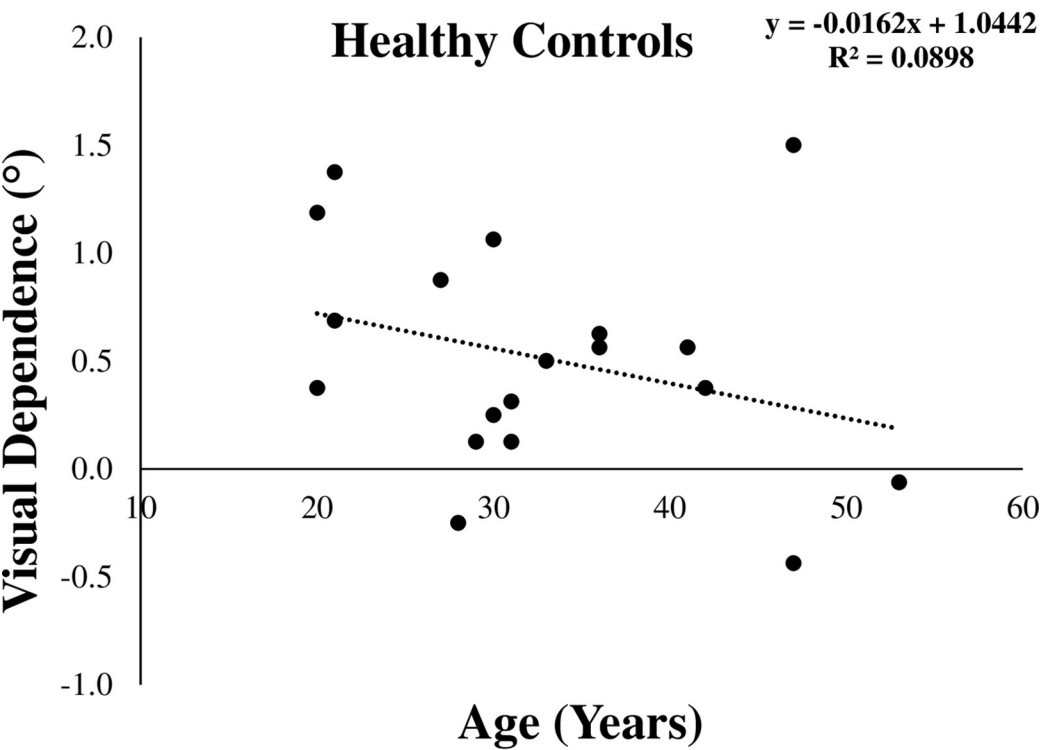

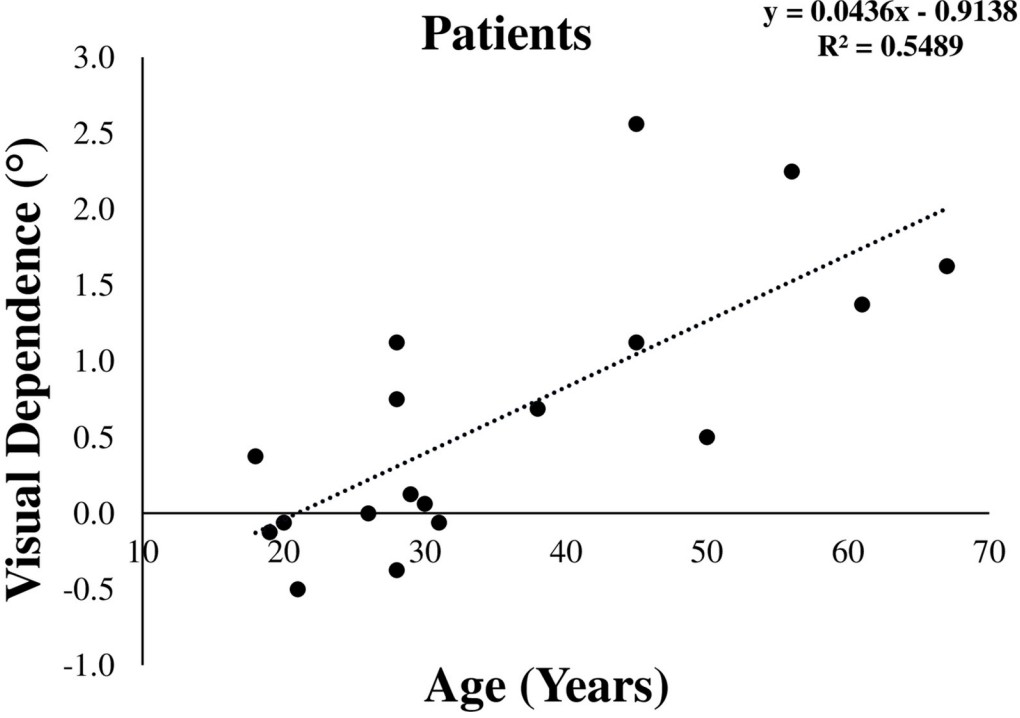

**Fig 4. Effect of age on visual dependence level in control and patient groups.** Negative values for VD indicate that static SVV was greater than dynamic SVV values.

vestibular information [6]. Nevertheless, the vestibular input has to be integrated with somato-sensory and especially visual information about vertical orientation of the three-dimensional space relative to the earth-centered gravitational force. As such, both studies suggest that patients with schizophrenia adequately combine self-generated vestibular cues and exteroceptive visual input to judge line verticality.

Numerous studies have consistently reported gender-related differences on visuospatial tests, with men usually performing better than women [24–27]. Similarly, there are studies confirming gender differences in visual dependence on the rod and frame test (RFT), which assesses the influence of the surrounding frame upon the accuracy of judging vertical alignment [28,29]. In the current study, there were no significant gender differences in visual dependence levels on the RDT. A possible explanation is that the percentage of females was only around 25% of the total cohort in each group, or alternatively, it is a genuine finding consistent with other studies reporting absence of gender-related effects on dynamic SVV or VD measured by the RDT in health and disease states [30,31].

Another parameter analysed in this study was the time for rod alignment in each trial. An interesting observation is the similarity in rod alignment time for static and dynamic SVV in both groups. This is quite surprising, as it would be expected that a rotating background disc would cause some disorientation, and it would take longer to adjust the rod to perceived vertical, in comparison to a stationary disc. It is unlike the RFT, in which presence of a tilted frame incurs longer alignment time than when the frame is not tilted [32], indicating that the temporal aspects of resolving conflicting visual information are not uniform across all tests of visual dependence.

The present study also investigated the relationship of VD level to symptom severity and illness duration. VD level in patients with schizophrenia did not correlate significantly with clinical severity measures, including general psychiatric symptoms, negative and positive psychiatric symptoms. This indicates that VD level in these patients is independent of symptom severity. However, the positive correlation between illness duration and VD level suggests that patients with longer duration of disease displayed greater visual dependence. It is not clear whether such an effect is a result of the older age of the patients with longer duration of illness, because in this group, age was significantly correlated with VD level, but this is probably not the case, as in the healthy control group, age was not correlated with VD level. There have been conflicting reports on the effect of ageing on visual dependence at the perceptual level, whether assessed with a rotating circular background (RDT) or a tilted stationary background as in the rod and frame test (RFT). Some studies reported increase in VD levels with ageing, attributing such an effect mainly to deterioration of the vestibular system. For instance, Kobayashi et al [33] reported robustness of the static SVV with ageing, whereas the dynamic SVV during rotation of a background scene gradually increased with age. Similarly, Alberts et al [34] reported that the bias in vertical perception by a tilted frame on the RFT and response variability become larger with increasing age and proposed an age-dependent shift towards visual dependence and down weighting of unreliable and noisy vestibular signals for perception of vertical.

The fact that our healthy control group did not exhibit increased VD levels with ageing is in agreement with previous findings. In a study establishing normative data for static and dynamic SVV in an Indian population, Ashish et al [30] reported for 82 healthy adult volunteers no significant difference between the age groups 20–40 years and 41–60 years in these

SVV measures. In the current study, the age of the healthy participants falls within this range, and most participants are considered young adults to middle-aged adults, with a maximum age of 53 years. The absence of effect of ageing in increasing VD levels in the control group supports the view that chronological age may not necessarily lead to increased visual dependence [35]. It is also possible that any age effects have not yet fully set in this age group, due to the progressive nature of the age effects on visual dependence [36]. The same relationship between age and VD would also be expected for the patient group as they were also in the same age range, with only two participants exceeding the age of 60 years. Accordingly, the observed increase in VD levels with age in the patient group may more likely relate to the development of schizophrenia-related pathology, rather than ageing *per se*.

The increased VD level with longer illness duration in patients with schizophrenia suggests that more chronically ill patients may weigh vestibular input at a lesser extent than less chronically ill patients for spatial orientation during dynamic visual conditions. This may have an influence on their ability to interact with a dynamic environment and may contribute to impairment of postural control. Whether subclinical vestibular-based sensory integrative dysfunctions are the root of such an effect in patients with chronic schizophrenia is not clear, but one cannot ignore the possible influence of antipsychotic drugs on vestibular function. Even though most patients in this study were on atypical antipsychotic drugs, there are reports on the effect of psychotics on vestibular and auditory function [37]. In a study on cats [38], the administration of chlorpromazine was reported to suppress the spontaneous as well as the increased vestibular neuronal firing following stimulation of the vestibular nerve. In a human study on a paediatric patient, chlorpromazine was reported to cause auditory disorders such as development of tinnitus as an adverse drug reaction [39].

## Conclusions

Limitations to this study include not directly measuring vestibular function in our participants and not obtaining the chlorpromazine equivalent doses of the antipsychotic drugs the patients were receiving for treatment. Despite this limitation, our results show robustness of verticality perception in patients with schizophrenia upon optokinetic stimulation, probably indicating to normal visual dependence levels for spatial orientation at a perceptual level. However. results point and to an increase in visual dependence levels with greater chronicity of the disease in these patients, probably indicating a subclinical vestibular impairment in these chronically ill patients.

## Supporting information

**S1 File. File of the minimal data set that supports the conclusions in this study.**
(XLSX)

## Acknowledgments

We thank the Ministry of Health in Bahrain for providing ethical approval and clearance for entering their facilities for data collection.

## Author Contributions

**Conceptualization:** Rima Abdul Razzak, Jeff Bagust.

**Data curation:** Haitham Jahrami, Maryam Ebrahim Ali.

**Formal analysis:** Rima Abdul Razzak.

**Investigation:** Rima Abdul Razzak.

**Methodology:** Rima Abdul Razzak.

**Project administration:** Rima Abdul Razzak.

**Resources:** Haitham Jahrami.

**Software:** Jeff Bagust.

**Supervision:** Rima Abdul Razzak, Haitham Jahrami.

**Validation:** Rima Abdul Razzak, Mariwan Husni.

**Writing – original draft:** Rima Abdul Razzak.

**Writing – review & editing:** Rima Abdul Razzak, Mariwan Husni.

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
