## [Decision Letter · Decision Letter 0]

21 Oct 2022

PONE-D-22-23565Perceptual Visual Dependence for Spatial Orientation in Patients with SchizophreniaPLOS ONE

Dear Dr. Razzak,

Thank you for submitting your manuscript to PLOS ONE. After careful consideration, we feel that it has merit but does not fully meet PLOS ONE’s publication criteria as it currently stands. Therefore, we invite you to submit a revised version of the manuscript that addresses the points raised during the review process. Two experts in the field have carefully reviewed the manuscript. Both reviewers aclnowledged the manuscript is well written with leaving some minor concerns as appended below.I will make the final decision after receipt of your reply and necessary revision.

We look forward to receiving your revised manuscript.

Kind regards,

Manabu Sakakibara, Ph.D.

Academic Editor

PLOS ONE

Journal Requirements:

Reviewers' comments:

Reviewer's Responses to Questions

**Comments to the Author**

1. Is the manuscript technically sound, and do the data support the conclusions?

Reviewer #1: Yes

Reviewer #2: Partly

2. Has the statistical analysis been performed appropriately and rigorously? 

Reviewer #1: Yes

Reviewer #2: Yes

3. Have the authors made all data underlying the findings in their manuscript fully available?

Reviewer #1: Yes

Reviewer #2: Yes

4. Is the manuscript presented in an intelligible fashion and written in standard English?

Reviewer #1: Yes

Reviewer #2: Yes

5. Review Comments to the Author

Reviewer #1: The manuscript is very well written and states a very useful finding about perceptual differences in individuals with psychosis. Every question/concern that came to mind while I was reading the manuscript was answered/addressed in the Discussion, so I have very few suggestions for improvement for this very excellent manuscript.

My two small suggestions are:

-- perhaps the statement "probably indicating a sub-clinical vestibular dysfunction" should be removed from the abstract, or changed to a statement like "consistent with previous reports of possible vestibular dysfunction in patients with schizophrenia" since there are no measurements of vestibular function in the present work

-- when I saw the data in Figure 3, I assumed (at first -- until I read the discussion) that this correlation was driven by age. The lack of correlation in controls is a key finding for supporting the conclusion that the correlation of SVV with disease duration is not simply because visual/vestibular integration is impaired with age. Perhaps the authors could include a 2nd panel in Fig. 3 showing the lack of correlation in the control population? This might be a very convincing visualization.

Reviewer #2: - RDT and VD explanations are well written. But I suggest the authors to group the first 3 paragraphs together for more coherence.

- Line 76 : The authors mention few articles on vestibular dysfunction in schizophrenia. The only one cited is from the 1970s. It would be wise to put one or two more recent references to support this argument.

- Line 83 : “This deficit may manifest as impaired functioning in dynamic visual environments.” Can you give some examples that are more meaningful?

- Line 112 : “All participants had normal or corrected-to-normal visual acuity.” Please specify the scale.

- Line 138 : “Subjects observed the screen while sitting upright on an armless chair”. What was the distance measured between the screen and the eyes of the patient? And why? Was this distance the same for everyone?

- Line 162 : “Participants were informed of the importance of spatial accuracy, and that the trials were not time restricted”. It would have been interesting to see how quickly the participants would have considered the task successful. Why didn't you measure this parameter?

- Line 279 : Regarding conflicting reports on the effect of ageing, don't you think that more recent articles have more legitimacy in the results? Also, did these articles use exactly the same task as yours? If no, how can you discuss with these results?

- The effect of gender was not discussed even though men represent nearly 75% of the sample. It is thus likely that the gender effect is significant. Was it included in your analyses?

- Do you have prospects for future research to address or possibly compensate for your limitations?

Comment to the editor : Although the research is interesting and innovative, the authors did not find significant results between patients and control group. I leave it to Plos One to validate or invalidate the article according to the coherence of the story and the results of the literature. However, I note that despite expectations contrary to the assumptions mentioned, the authors were able to adequately discuss the results found in relation to the literature.

6. PLOS authors have the option to publish the peer review history of their article (what does this mean?). If published, this will include your full peer review and any attached files.

Reviewer #1: **Yes: **Cheryl Olman

Reviewer #2: No

---

## [Author Response · Author response to Decision Letter 0]

25 Oct 2022

Response to Editor 

The authors confirm that the submitted manuscript meets PLOS ONE’s requirements including those for file naming.

We have included the following paragraph in the manuscript detailing how informed consent was obtained: “After the aim and impact of the study were explained to the healthy controls and to the patients and their relatives, informed verbal consent was taken. An impartial witness, who was not a member of the study team and worked at the center where the patients were treated, endorsed that the consent from patients and their relatives was voluntary and freely given. Only those who volunteered were included in the study”.

3. In your Data Availability statement, you have not specified where the minimal data set underlying the results described in your manuscript can be found. PLOS defines a study's minimal data set as the underlying data used to reach the conclusions drawn in the manuscript and any additional data required to replicate the reported study findings in their entirety. All PLOS journals require that the minimal data set be made fully available. 

The minimal data set can be found in an Excel file uploaded as Supporting Information File. 

There were no citations of papers that have been retracted. However additional citations were included to further support the Discussion section.

Response to Reviewers

The authors would like to thank both reviewers for their comments which enhanced the quality of the manuscript. The authors accommodated all comments as best as they can.

Reviewer #1: 

The manuscript is very well written and states a very useful finding about perceptual differences in individuals with psychosis. Every question/concern that came to mind while I was reading the manuscript was answered/addressed in the Discussion, so I have very few suggestions for improvement for this very excellent manuscript.

My two small suggestions are:

-- Perhaps the statement "probably indicating a sub-clinical vestibular dysfunction" should be removed from the abstract, or changed to a statement like "consistent with previous reports of possible vestibular dysfunction in patients with schizophrenia" since there are no measurements of vestibular function in the present work.

We took the reviewer’s suggestion and changed the statement according to their recommendation.

-- when I saw the data in Figure 3, I assumed (at first -- until I read the discussion) that this correlation was driven by age. The lack of correlation in controls is a key finding for supporting the conclusion that the correlation of SVV with disease duration is not simply because visual/vestibular integration is impaired with age. Perhaps the authors could include a 2nd panel in Fig. 3 showing the lack of correlation in the control population? This might be a very convincing visualization.

We included an additional figure depicting the effect of ageing on visual dependence in both healthy controls and patients (2 graphs).

Reviewer #2: 

- RDT and VD explanations are well written. But I suggest the authors to group the first 3 paragraphs together for more coherence.

The authors thank the reviewer for this suggestion, but they felt that it was sufficient to group the first two paragraphs only, so as not to make it into a very long paragraph.

- Line 76: The authors mention few articles on vestibular dysfunction in schizophrenia. The only one cited is from the 1970s. It would be wise to put one or two more recent references to support this argument.

The authors added a relatively more recent reference: “Haghgooie S, Lithgow B J, Gurvich C, Kulkarni J. Quantitative detection and assessment of schizophrenia using electrovestibulography. 4th International IEEE/EMBS Conference on Neural Engineering, 2009; 486 – 489. doi: 10.1109/NER.2009.5109339.

- Line 83: “This deficit may manifest as impaired functioning in dynamic visual environments.” Can you give some examples that are more meaningful?

The authors provided some real-life examples such as crowded or busy environments, as walking in supermarket aisles, movements of crowds or traffic, moving images at the cinema, trees swaying, or while driving fast on a highway.

- Line 112: “All participants had normal or corrected-to-normal visual acuity.” Please specify the scale.

The authors ensured that the participants had 20/20 visual acuity, and participants were allowed to put on their corrective spectacles if necessary. This was added in the Methods section.

- Line 138: “Subjects observed the screen while sitting upright on an armless chair”. What was the distance measured between the screen and the eyes of the patient? And why? Was this distance the same for everyone?

According to many studies on RFT and RDT, the distance between the screen and the eyes of the participant is 80 cm. The center of the screen was at eye level of the observer. This was added in the Methods section. Yes, the distance was the same for all participants, as they were all tested on the same computer/screen set up and the same place in the Psychiatry hospital.

- Line 162: “Participants were informed of the importance of spatial accuracy, and that the trials were not time restricted”. It would have been interesting to see how quickly the participants would have considered the task successful. Why didn't you measure this parameter?

The authors thank the reviewer for pointing this out. Fortunately, the software does record and stores the time of rod alignment for each trial. The authors analysed the time-related data and included the findings in the Results section. The significance of these results was included in the Discussion.

- Line 279: Regarding conflicting reports on the effect of ageing, don't you think that more recent articles have more legitimacy in the results? Also, did these articles use exactly the same task as yours? If no, how can you discuss with these results?

The authors did add more recent references in the Discussion on the effect of ageing on visual dependence, especially those utilizing the common and similar tests of visual dependence, specifically the RFT and the RDT.

- The effect of gender was not discussed even though men represent nearly 75% of the sample. It is thus likely that the gender effect is significant. Was it included in your analyses?

The authors appreciate this recommendation from the reviewer. Upon their suggestion, gender analysis was included, and the findings were addressed, with added references to enrich the Discussion.

- Do you have prospects for future research to address or possibly compensate for your limitations?

Definitely. The authors would like in future studies to add assessment of peripheral vestibular system function in patients with long-duration schizophrenia and determine any possible interference of long-term use of neuroleptic medications on vestibular functions.

---

## [Decision Letter · Decision Letter 1]

18 Nov 2022

PONE-D-22-23565R1Perceptual Visual Dependence for Spatial Orientation in Patients with Schizophrenia

PLOS ONE

Dear Dr. Razzak,

Thank you for submitting your manuscript to PLOS ONE. After careful consideration, we feel that it has merit but does not fully meet PLOS ONE’s publication criteria as it currently stands. Therefore, we invite you to submit a revised version of the manuscript that addresses the points raised during the review process.

Two original reviewers have carefully reviewed the revision. Their comments are appended below. Both reviewers are satisfied with your response and revision. 

I judged the revised manuscript is potentially acceptable with leaving some minor concerns which will strengthen your study. Please consider these remarks. 

We look forward to receiving your revised manuscript.

Kind regards,

Manabu Sakakibara, Ph.D.

Academic Editor

PLOS ONE

Journal Requirements:

Reviewers' comments:

Reviewer's Responses to Questions

**Comments to the Author**

1. If the authors have adequately addressed your comments raised in a previous round of review and you feel that this manuscript is now acceptable for publication, you may indicate that here to bypass the “Comments to the Author” section, enter your conflict of interest statement in the “Confidential to Editor” section, and submit your "Accept" recommendation.

Reviewer #1: All comments have been addressed

Reviewer #2: All comments have been addressed

2. Is the manuscript technically sound, and do the data support the conclusions?

Reviewer #1: Yes

Reviewer #2: Yes

3. Has the statistical analysis been performed appropriately and rigorously? 

Reviewer #1: Yes

Reviewer #2: Yes

4. Have the authors made all data underlying the findings in their manuscript fully available?

Reviewer #1: Yes

Reviewer #2: Yes

5. Is the manuscript presented in an intelligible fashion and written in standard English?

Reviewer #1: Yes

Reviewer #2: Yes

6. Review Comments to the Author

Reviewer #1: My few concerns from the previous round of review have been addressed. On re-reading the abstract I noticed that on the first use of SVV when it is defined it is defined as perceived visual vertical instead of subjective visual vertical, which might be mildly puzzling to readers; just a tiny change you might think about making

Reviewer #2: Dear authors,

I take into account the fact that you carried out a real work of bibliography following my remarks on the manuscript. I consider that your work to date is well done, in particular with the addition of recent bibliographic research as well as concrete examples illustrating your words. I transmitted to PLOS ONE the acceptance of the manuscript.

These last comments will help you to finalize your manuscript =

- Line 113 = I think I was misunderstood. I expect that the participants have normal or corrected-to-normal visual acuity. However, it would be wise to clarify the name of the scale used to measure the visual acuity. Is it a Snellen chart? Is it a Monoyer scale? Please clarify. I think this is an important criterion if other authors would like to follow your methodology.

- Line 139 = Indeed, most visual studies mention eye-screen distances between 40cm and 100cm depending on the projection of the different stimuli. In relation to your biographical research, your answer is judicious.

- Line 328 = I see that my point about age and gender effect has been perfectly addressed in the manuscript, with solid argumentation based on recent literature. This is a very good point.

- Line 344 = Similarly, the remark about the time of rod alignment for each trial and also the time-related data were added in the results as well as the discussion and confronted with the literature. However, although the results were not significant between the two groups, I do not think it is necessary to comment on this.

- Line 351 and line 369 = I find that the 2 paragraphs dealing with the effect of age on visual dependence are particularly well discussed. You compare the results currently reported in the article with those in the literature, and it's great.

- Line 372 = Finally, it would have been wise to specify the number of people recruited in the study of Ashish. I also appreciated that you compared it with your current article by indicating age ranges and specifying the age range of your population.

7. PLOS authors have the option to publish the peer review history of their article (what does this mean?). If published, this will include your full peer review and any attached files.

Reviewer #1: No

Reviewer #2: No

---

## [Author Response · Author response to Decision Letter 1]

18 Nov 2022

Reviewer #1: 

My few concerns from the previous round of review have been addressed. On re-reading the abstract I noticed that on the first use of SVV when it is defined it is defined as perceived visual vertical instead of subjective visual vertical, which might be mildly puzzling to readers; just a tiny change you might think about making

We have substituted the word “perceived” by “subjective” in the abstract.

Reviewer #2: Dear authors,

I take into account the fact that you carried out a real work of bibliography following my remarks on the manuscript. I consider that your work to date is well done, in particular with the addition of recent bibliographic research as well as concrete examples illustrating your words. I transmitted to PLOS ONE the acceptance of the manuscript.

We, the authors, would like to thank the reviewer for their encouragement and confidence in the changes made in the first revision of the manuscript.

These last comments will help you to finalize your manuscript =

- Line 113 = I think I was misunderstood. I expect that the participants have normal or corrected-to-normal visual acuity. However, it would be wise to clarify the name of the scale used to measure the visual acuity. Is it a Snellen chart? Is it a Monoyer scale? Please clarify. I think this is an important criterion if other authors would like to follow your methodology.

We did add that the scale used to measure the visual acuity is the Snellen chart.

- Line 139 = Indeed, most visual studies mention eye-screen distances between 40cm and 100cm depending on the projection of the different stimuli. In relation to your biographical research, your answer is judicious.

We thank the reviewer for this encouraging comment.

- Line 328 = I see that my point about age and gender effect has been perfectly addressed in the manuscript, with solid argumentation based on recent literature. This is a very good point.

We thank the reviewer for this encouraging comment.

- Line 344 = Similarly, the remark about the time of rod alignment for each trial and also the time-related data were added in the results as well as the discussion and confronted with the literature. However, although the results were not significant between the two groups, I do not think it is necessary to comment on this.

We did remove from the Discussion section the paragraph where we attempted to provide a reason for the absence of group difference in time of rod alignment.

- Line 351 and line 369 = I find that the 2 paragraphs dealing with the effect of age on visual dependence are particularly well discussed. You compare the results currently reported in the article with those in the literature, and it's great.

We thank the reviewer for this encouraging comment.

- Line 372 = Finally, it would have been wise to specify the number of people recruited in the study of Ashish. I also appreciated that you compared it with your current article by indicating age ranges and specifying the age range of your population.

We included the number of people recruited (82 healthy adult volunteers) in the study of Ashish.

---

## [Editor Report · Decision Letter 2]

22 Nov 2022

Perceptual Visual Dependence for Spatial Orientation in Patients with Schizophrenia

PONE-D-22-23565R2

Dear Dr. Razzak,

We’re pleased to inform you that your manuscript has been judged scientifically suitable for publication and will be formally accepted for publication once it meets all outstanding technical requirements.

Kind regards,

Manabu Sakakibara, Ph.D.

Academic Editor

PLOS ONE
---

## [Editor Report · Acceptance letter]

24 Nov 2022

PONE-D-22-23565R2 

Perceptual visual dependence for spatial orientation in
patients with schizophrenia 

Dear Dr. Razzak:

I'm pleased to inform you that your manuscript has been deemed suitable for publication in PLOS ONE. Congratulations! Your manuscript is now with our production department. 

Kind regards, 

on behalf of

Dr. Manabu Sakakibara 

Academic Editor

PLOS ONE